# Leveraging the True Depth of LLMs

**Ramón Calvo González**                                                  *ramon.calvogonzalez@unige.ch*
*University of Geneva*

**Daniele Paliotta**
*University of Geneva*

**Mattéo Pagliardini**
*EPFL*

**Martin Jaggi**
*EPFL*

**François Fleuret**
*University of Geneva*
*FAIR at META*

**Reviewed on OpenReview:** *https://openreview.net/forum?id=JccJ6YfWd4*

## Abstract

The remarkable capabilities of Large Language Models (LLMs) are overshadowed by their immense computational cost. While recent work has shown that many LLM layers can be reordered or even removed with minimal impact on accuracy, these insights have not been translated into significant inference speedups. To bridge this gap, we introduce a novel method that restructures the computational graph by grouping and evaluating consecutive layer pairs in parallel. This approach, requiring no retraining, yields a 1.19x throughput gain on Llama 2 7B while reducing the average benchmark accuracy by only 1.5%. We demonstrate the practical value of this method for large-scale LLM deployment and show that some of the lost accuracy can be recovered with lightweight fine-tuning of the parallelized layers.

## 1 Introduction

The rapid advancement of LLMs has revolutionized Artificial Intelligence applications across industries. However, the ever-increasing computational demands of these models, with parameters often numbering hundreds of billions, present significant commercial challenges. Efficient inference is crucial for organizations that deploy these models at scale, as it directly impacts operational costs, user experience, and environmental sustainability (Singh et al., 2025; Xu et al., 2024; Wu et al., 2022). Monthly cloud computing expenses for LLM inference can reach millions of dollars for high-traffic applications, making optimization techniques essential. In addition, reducing inference latency is critical for real-time applications and for deploying models on devices with more limited compute.

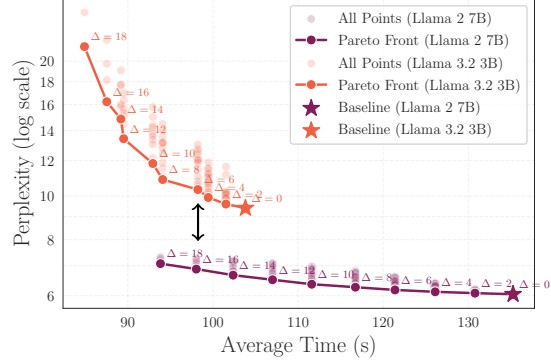

Figure 1: The effect of LP on execution time (4K tokens) and perplexity (measured against RedPajama (Together Computer, 2023)).

Thus, the development and implementation of efficient inference methods has become a key differentiator in the competitive AI market, driving both innovation and profitability.

LLMs have evolved to incorporate architectures with hundreds of layers (OpenAI, 2023; authors, 2024). These models are constructed from stacked transformer blocks, each comprising attention and feedforward subblocks, with a residual stream traversing the entire architecture to facilitate efficient gradient propagation during training. This architectural choice parallels the design principles of ResNets (He et al., 2015), where research has shown that network depth may be partially redundant, allowing layer reordering without significant performance loss (Veit et al., 2016). Recent investigations have revealed similar flexibility in transformer architectures (Lad et al., 2024), where interventions such as layer removal and swapping are applied without large performance degradations. Although these findings challenge our understanding of LLMs' true effective depth, their potential for optimizing inference efficiency remains unexplored.

Inspired by this observed layer independence, we investigated several interventions to the computational graph of pre-trained LLMs. Our exploration of layer shuffling,

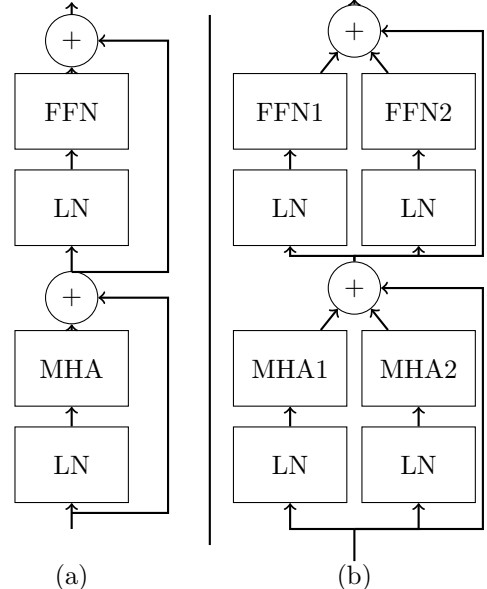

(a)          (b)

Figure 2: Comparison of a normal transformer block (a) with our layer parallel implementation (b). Divergent paths in (b) are split across the Tensor Parallel axis (Eq. LP).

pruning, and merging revealed that multiple consecutive block pairs can be processed in parallel while maintaining accuracy across perplexity and In-Context Learning (ICL) benchmarks. This led us to propose Layer Parallelism (LP), a novel approach that enhances inference speed when performing inference in the Tensor Parallel (TP) regime. LP modifies the computational graph of a pre-trained LLM to reduce the inter-device communication by half, with a minimal drop in model performance. Furthermore, we show that this performance degradation can be partially mitigated through targeted fine-tuning procedures.

**Contributions.** Our contributions can be summarized as follows:

- We explore the space of interventions on the layers of a pre-trained LLM and find that some transformations, such as contiguous parallelization, preserve model performance.

- We find that we can define a parallelization transform on the computational graph of two sequential Transformer layers, and stack this parallelization operation across several sequential pairs of layers without losing significant ICL performance. Our approach, which we call LP, can be applied to existing Transformer models.

- We show that by fine-tuning the LP blocks we can recover some of the lost performance, while retaining the previously obtained speed-up.

## 2 Related work

**The effective depth of Deep Networks.** Theoretically, given enough width, any feed-forward network with at least one hidden layer can model any function (Pinkus, 1999). In practice, it is easier to achieve high expressivity by increasing the model's depth. However, naively increasing network depth can complicate optimization, since the gradients now have to flow through many layers. To alleviate this problem, ResNets (He et al., 2015) introduced skip connections at regular intervals to allow an easy flow of the gradient to the first layers. Alternatively, Inception (Szegedy et al., 2014) explored approaches to boost computational power by adding additional processing units along different parallel pathways in the computational network, rather than just along a single sequential path. A unification of both methods can be found in the Highway

Networks (Srivastava et al., 2015), where the skip connection of the residual blocks consists of another block of compute. Nowadays, residual connections are ubiquitous in large models.

**Efficient inference of LLMs.** Several complementary approaches exist for enhancing the computational efficiency of large-scale models, primarily through pruning/sparsity, quantization, and parallelism. Pruning (LeCun et al., 1989; Han et al., 2015; 2016; Frantar & Alistarh, 2023) constitutes a dimensional reduction methodology that systematically eliminates redundant parameters while preserving model performance, thereby introducing architectural sparsity. This methodology is founded on empirical evidence demonstrating that neural networks frequently exhibit overparameterization, containing numerous weights with negligible contributions to the output. Through sophisticated pruning strategies, the inherent sparsity support in contemporary accelerators can be leveraged to enhance both memory utilization and computational efficiency (Zhang et al., 2020; Wang et al., 2021). Early-exit (Teerapittayanon et al., 2017; Zhou et al., 2020) can be seen as a way of runtime layer-wise pruning, which halts the LLM forward pass when the next token certainty is high in the intermediate layers. This approach can also be thought of as a way of reducing the effective depth of the model at test time. In contrast, quantization encompasses the transformation of floating-point numerical representations (predominantly FP32) into reduced-precision integer formats, such as INT8 or INT4 (Shen et al., 2019; Han et al., 2016; Jacob et al., 2018). When implemented on hardware accelerators, these lower-precision representations allow for higher FLOPs and better use of the memory bandwidth, addressing a primary bottleneck in modern large-scale models (Gholami et al., 2024); moreover, integer-based computations yield enhanced processing speed and substantially improved energy efficiency (Horowitz, 2014). Finally, parallelization techniques during inference, such as tensor and pipeline parallelism, enable the distribution of computational workload across multiple accelerators, thereby reducing latency and increasing throughput, although this often requires careful consideration of communication overhead and load balancing (Li et al., 2024; Narayanan et al., 2021).

**Tensor-parallel optimizations.** Inter-device communication remains the dominant bottleneck in tensor parallelism, as each transformer sub-module imposes a synchronization step. Recent work targets this cost by either removing synchronization points or shrinking the communicated tensors. Sync-Point Drop (Kim et al., 2025) cuts the number of all-reduce operations by modifying block structure so that local attention outputs propagate without immediate aggregation; layers are ranked by sensitivity to synchronization removal, and only the sensitive ones receive targeted tuning, yielding a 20% speedup with roughly 1% accuracy loss. A complementary direction reduces communication volume. (Dong et al., 2024) quantizes per-layer activation exchanges, reaching a $3.8\times$ compression ratio with about a 2% drop on the evaluated benchmarks.

**Parallel attention-feedforward fusion.** GPT-J (Wang & Komatsuzaki, 2021) introduced a parallel formulation of the transformer decoder layer, executing attention and feedforward sub-blocks concurrently:

$$y = x + \mathrm{MHA}(\mathrm{LN}_{\mathrm{MHA}}(x)) + \mathrm{FFN}(\mathrm{LN}_{\mathrm{FFN}}(x))$$

For models trained with tensor parallelism, this modification halves the number of required all-reduce operations. Additionally, it reduces memory bandwidth usage by eliminating one read and write operation of hidden states from HBM. The input projections for the attention and MLP sub-blocks can also be fused into a single kernel, further increasing arithmetic density. Consequently, training time is reduced by approximately 15% without observable performance degradation. PaLM (Chowdhery et al., 2023) also adopted this formulation, noting that negative effects from deviating from the standard transformer self-attention diminish with increasing model size. This parallel formulation continues to be employed in more recent LLMs, including Gemini 1.5 Flash (Georgiev et al., 2024).

In contrast to these methods, which require training a new model from scratch with a modified architecture, our approach is applied post-hoc to already-trained models (authors, 2023; 2024; Yang et al., 2025). This highlights two other key differences. First, the granularity of parallelism differs: GPT-J-style models parallelize the attention and feed-forward sub-blocks *within* a single layer, whereas our method parallelizes *entire consecutive layers*, directly reducing the model's effective depth. Second, LP accepts a trade-off by approximating the original computation, which results in a slight performance degradation in exchange for inference acceleration. This degradation can be largely recovered with light fine-tuning. The GPT-J architecture, by contrast, is exact by definition, as the model was trained with it from the beginning.

**Parallelism via Computational Graph Optimization.** Recent research has investigated architectural layer-level optimization strategies to enhance transformer model inference efficiency. The Staircase Transformer (Cutler et al., 2025) implements parallel layer execution with dynamic recurrent computation based on model requirements. Similarly, the Staggering Transformer (Cai et al., 2024) achieves layer parallelization by connecting layer $l_k$ at time step $t$ to both the $(t-1)$ output of layer $l_{k-1}$ and the $t$ output of layer $l_{k-2}$. To the best of our knowledge, no research has addressed the fusion of consecutive layers through tensor parallelism.

## 3 Effective Depth

In this section we investigate the effective depth of pretrained LLMs by applying several transformations and measuring the resulting perplexity degradation. We reveal loose dependencies between intermediate layers. The transformations consist of shuffling, merging, and pruning transformer layers. To avoid the combinatorial explosion resulting from considering all possible subsets of transformer layers, we instead apply our transformations to all contiguous stretches of layers. If $L = \{\ell_1, \ldots, \ell_N\}$ are the ordered layers, then we apply our transformations to all the sublists $\{\ell_i\}_{i=s}^e$ with $1 \leq s \leq e \leq N$. Previous works have shown that—at least when considering pruning—the importance of layers is well-behaved, with low-importance layers close to one another (Men et al., 2024), which justifies considering contiguous stretches of layers only.

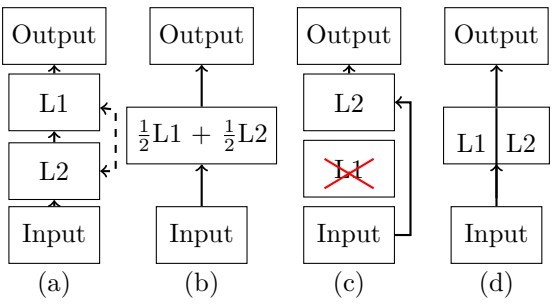

Figure 3: **Diagram of transformations applied in § 3.** Diagrams **(a,b,c,d)** represent shuffling, merging, pruning and parallel respectively.

**Shuffling, pruning and merging blocks.** We start by investigating the effect of several transformations on the model's perplexity. First, we experiment with shuffling contiguous stretches of layers (Fig. 3a), reordering them according to random permutations. Results, shown in Fig. 4(a), reveal that while shuffling the early and late layers is detrimental, there are large stretches of intermediate blocks that can be shuffled with surprisingly low impact on perplexity. For instance, one can shuffle layers 15 through 24 of Llama 2 7B and only increase perplexity by 2.9. This suggests that many layers may operate at a similar level of abstraction, challenging the classical belief of strictly hierarchical representations. This observed layer decoupling is a key insight. We also experiment with pruning (Fig. 3c) and merging (Fig. 3b) contiguous layers. Pruning, studied in prior works (Gromov et al., 2024; Jung et al., 2019), involves removing layers entirely. Merging involves averaging the weights of consecutive layers. We find that both transformations lead to a more significant perplexity increase compared to shuffling (see Fig. 4b and Fig. 4c). Merging, in particular, offers no advantage over pruning, suggesting that naively combining weights from different layers is ineffective. These initial experiments indicate that while layers are robust to reordering, their individual parameters are crucial.

**Running blocks in parallel.** The observed layer decoupling suggests that specific transformer operations may be executed independently, providing an opportunity for parallel computation. More precisely, let's consider two sequential transformer layers $\ell_k$ and $\ell_{k+1}$, each comprising attention and Feed-Forward Network (FFN) sub-blocks ($A_k(\cdot)$ and $F_k(\cdot)$, respectively). The standard sequential output $\boldsymbol{y}$ for these layers, given an input $\boldsymbol{x}$, is given by:

$$
\begin{aligned}
\boldsymbol{y} = \boldsymbol{x} &+ A_k(\boldsymbol{x}) \\
&+ F_k(\boldsymbol{x} + A_k(\boldsymbol{x})) \\
&+ A_{k+1}(\boldsymbol{x} + A_k(\boldsymbol{x}) + F_k(\boldsymbol{x} + A_k(\boldsymbol{x}))) \\
&+ F_{k+1}(\boldsymbol{x} + A_k(\boldsymbol{x}) + F_k(\boldsymbol{x} + A_k(\boldsymbol{x}))) \\
&+ A_{k+1}(\boldsymbol{x} + A_k(\boldsymbol{x}) + F_k(\boldsymbol{x} + A_k(\boldsymbol{x}))))
\end{aligned}
\tag{1}
$$

$$\text{(SEQ)}$$

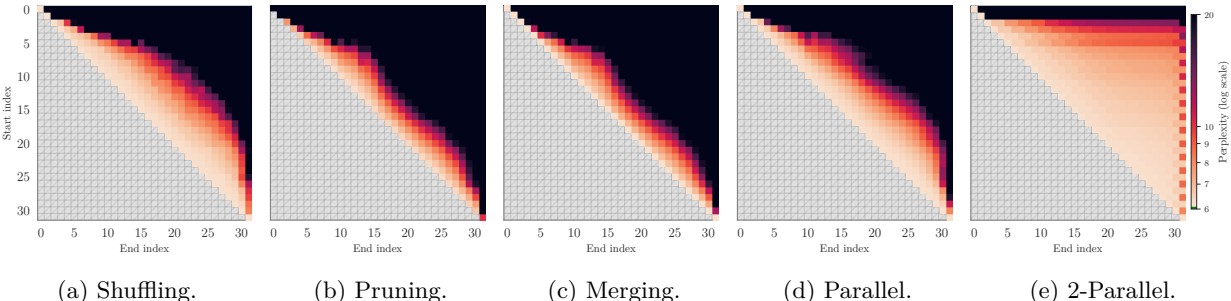

| (a) Shuffling. | (b) Pruning. | (c) Merging. | (d) Parallel. | (e) 2-Parallel. |

Figure 4: **Changes in perplexity when applying transformations on contiguous stretches of layers.** Each of the five heatmaps above corresponds to a transformation of a group of consecutive layers, where the row index $s$ corresponds to the first layer of the group, and the column index $e$ to the last. The color coding indicates how the perplexity—estimated on a subset of RedPajama (Together Computer, 2023)—is impacted by the corresponding modification of the model. The perplexity for the base Llama 2 7B model is 6.2. In **(a)**, we shuffle—for each forward—the layers from $s$ to $e$. We can see that many consecutive layers can be shuffled with little impact on the overall perplexity. For instance, shuffling layers 15 to 25—10 layers in total—raises the perplexity only to 9.1. In **(b)**, we prune contiguous stretches of layers. We can see that not many blocks can be removed without starting to significantly degrade the perplexity. In **(c)** we merge contiguous layers. The results with merging are nearly identical to those for pruning. This reveals there is no advantage in merging layers, most likely a result of averaging matrices that originate from different initial values. In **(d)** we run contiguous blocks in parallel. Given the success of shuffling, it makes sense that this approach works well. Running blocks 17 to 27 raises the perplexity to 9.3. Finally, in **(e)** we run *pairs of consecutive layers* in parallel. As a result, we can parallelize much longer stretches of layers. For instance, we can apply this transformation from layer 4 to 29 and only increase the perplexity to 9.1. This reduces the depth of the model from 32 to 19. This result makes it possible for us to leverage this parallelism for faster inference as we discuss in § 4.

The highlighted terms represent the first block's contribution to the second block's processing. Given the observed layer independence, we can hypothesize that these terms have minimal impact, leading to the following approximation:

$$\hat{\boldsymbol{y}} = \boldsymbol{x} + A_k(\boldsymbol{x}) + F_k(\boldsymbol{x} + A_k(\boldsymbol{x})) + A_{k+1}(\boldsymbol{x}) + F_{k+1}(\boldsymbol{x} + A_{k+1}(\boldsymbol{x})) \quad\quad\quad \text{(PAR)}$$
$$\approx \boldsymbol{x} + A_k(\boldsymbol{x}) + A_{k+1}(\boldsymbol{x}) + F_k(\boldsymbol{x} + A_k(\boldsymbol{x}) + A_{k+1}(\boldsymbol{x})) + F_{k+1}(\boldsymbol{x} + A_k(\boldsymbol{x}) + A_{k+1}(\boldsymbol{x})) \quad \text{(LP)}$$

This approximation enables parallel execution of blocks $\ell_k$ and $\ell_{k+1}$ through divergent computational paths. We experiment with running contiguous stretches of layers in parallel and show our results in Fig. 4d. We observe results similar to shuffling. Unlike shuffling, this approach allows us to potentially improve the runtime through enhanced parallelism. We show how we can, for instance, run layers 17 to 27 in parallel, only losing 3.1 perplexity points, while reducing the depth of the model from 32 to 23.

To assess how strongly attention and FFN sub-blocks rely on the residual stream generated by preceding layers, we apply a CKA-based comparison (Kornblith et al., 2019). For each prompt drawn from a small RedPajama subset, we record module activations under two conditions: a standard forward pass and a counterfactual pass in which the incoming residual contribution is removed before processing the next block. The standard decoder update for layer $k$ is:

$$h_k = x + A_k(x) + F_k\big(x + A_k(x)\big)$$
$$A_{k+1} = A_{k+1}(h_k)$$
$$F_{k+1} = F_{k+1}\big(h_k + A_{k+1}\big)$$
$$h_{k+1} = h_k + A_{k+1} + F_{k+1}$$

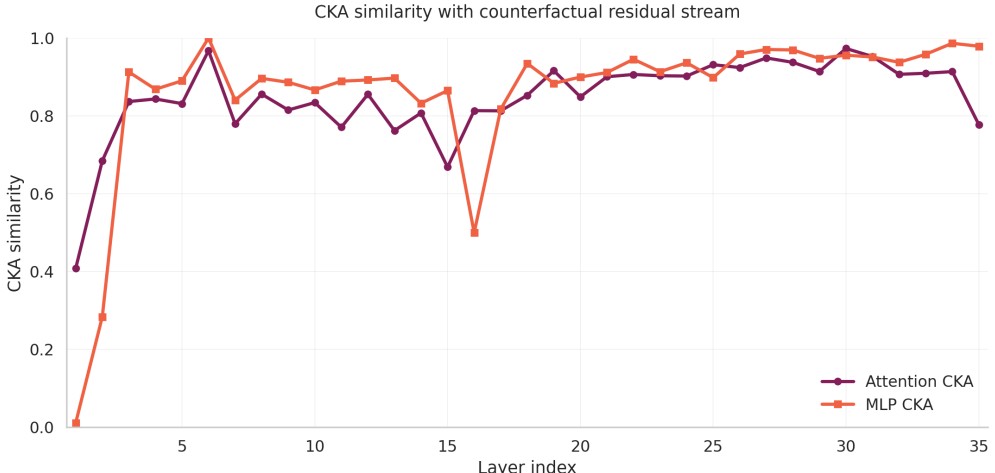

Figure 5: CKA similarity for Qwen3-4B between the original MHA/FFN activations and the counterfactual activations that exclude incoming residuals. Higher values imply greater invariance to the upstream residual stream. A plateau of high CKA similarity between pairs of layers is preceded by a sharp similarity decline at layer 16, which coincides with the performance degradations experienced when applying different levels of LP at different positions.

Counterfactual activations are obtained by subtracting the residual stream before evaluating the next block:

$$\tilde{h}_k = h_k - A_k(x) - F_k(x + A_k(x))$$
$$\tilde{A}_{k+1} = A_{k+1}(\tilde{h}_k)$$
$$\tilde{F}_{k+1} = F_{k+1}(\tilde{h}_k + \tilde{A}_{k+1})$$

Similarity between original and counterfactual activations is then computed via CKA:

$$S_A^k = \mathrm{CKA}(A_k, \tilde{A}_k)$$
$$S_F^k = \mathrm{CKA}(F_k, \tilde{F}_k)$$

As shown in Fig. 5, layers that tolerated the 2-parallel intervention also show high CKA similarity, indicating limited dependence of their attention and FFN computations on the immediate residual input.

**Contiguous 2-parallel.** Instead of parallelizing long stretches of layers, we experiment with running *pairs of consecutive layers* in parallel. This springs from the assumption that local ordering matters less than global ordering, i.e. shuffling consecutive layers introduces fewer potential issues than shuffling layers separated by larger distances. As an example, if we apply the proposed transformation to layers $\{\ell_{15}, \ell_{16}, \ell_{17}, \ell_{18}, \ell_{19}\}$, it would result in the following process: (1) the two layers $\{\ell_{15}, \ell_{16}\}$ process the input in parallel (according to equation (PAR)), (2) the output is forwarded to layers $\{\ell_{17}, \ell_{18}\}$ which process it in parallel; finally, in (3) their joint output is fed to layer $\ell_{19}$ which processes it on its own as any normal layer. The effect of such a transformation on the compute graph can be seen in Fig. 4e. Remarkably, it is possible to run wide stretches of consecutive pairs of blocks in parallel with only a minor degradation of perplexity. For instance, one can apply this transformation from layer 4 to layer 29 with only a perplexity increase of 2.9, while reducing the model depth from 32 to 19. The success of this approach led us to also try running triplets of consecutive layers in parallel, but we found it to perform worse.

## 4 Efficient Parallelization of Blocks

Naively trying to fuse two attention or MLP sub-blocks does not result in a noticeable improvement of inference speed for large batch sizes and sequence lengths, since in these situations, inference approaches

the the compute-bound regime. For this reason we focus our attention on the Tensor Parallel setting, where each module's weights are split over two or more GPUs. Rearranging the computational graph of two contiguous layers like in Fig. 2b effectively reduces the number of inter-GPU synchronizations by half. Now, each divergent path is computed in parallel over multiple accelerators, and only the single intermediate and final results need to be synchronized. While this approach is not numerically equivalent to (PAR), we nonetheless—and quite surprisingly— show that it works well on already trained models, circumventing the need to train from scratch.

**LP Multi-Head Attention.** Traditional tensor parallelism in MHA distributes attention heads evenly across GPUs (Shoeybi et al., 2020), performing self-attention and output projection locally before gathering results through an all-reduce summation. Each GPU processes tensors of dimensions $Q, K, V, att \in \mathbb{R}^{T \times \frac{D}{g}}$, where $T$ is sequence length, $D$ is feature dimension, and $g$ is the number of parallel workers. The local output projection produces a low-rank $o_i \in \mathbb{R}^{T \times D}$ for each worker $i$, and then a final all-reduce operation performs $o = \sum_i^g o_i$, computing the final output.

To implement LP, we increase the depth of the query, key, and value weight matrices ($W_Q, W_K, W_V \in \mathbb{R}^{(g_n \cdot h_d) \times D}$) and widen the output projection ($W_O \in \mathbb{R}^{D \times (n_h \cdot h_d)}$), where $n_h$ represents heads per GPU and $h_d$ is head dimensionality. The reduction operation now will simultaneously compute the full-rank output projections and the sum of all parallel layers (Fig. 6). Note that this approach requires that $n_h$ is divisible by the total number of GPUs.

**LP FFN.** Standard tensor parallelism for single-hidden-layer FFNs splits the first layer's output across devices, generates low-rank outputs from the second layer, and sums them through reduction. To parallelize two FFN layers, we double the first layer's output dimensionality and perform separate output (low-rank) projections for each layer. A single reduction operation then serves the dual purpose of computing full outputs for each layer and combining their results, as shown in Fig. 2(b). In summary, LP for FFN just concatenates the up-projection weights and continues as normal TP, allowing for multiple GPUs to be allocated per parallelized layer.

**Handling of the LayerNorms.** Since we assume at least one GPU per parallelized layer, we can assign each original LayerNorm to the divergent path that contains the attention and FFN blocks from its original layer. We have also observed that using the same merged LayerNorm with linear interpolation and spherical linear interpolation in each divergent path yields good results. For the sake of simplicity, we conduct all of our experiments using the original LayerNorms on each divergent path.

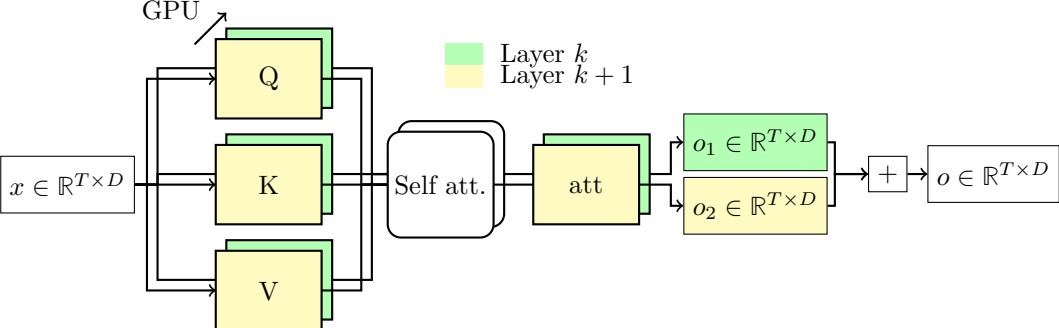

Figure 6: **LP attention implementation**. This diagram shows the implementation of the LP attention from Fig. 2b. The stacked layers represent different GPUs, the colors indicate different layers and the arrows express linear projections. In this case, the number of GPUs and the number of parallelized layers coincides and is two, which is the set-up that we use for all our experiments in this work.

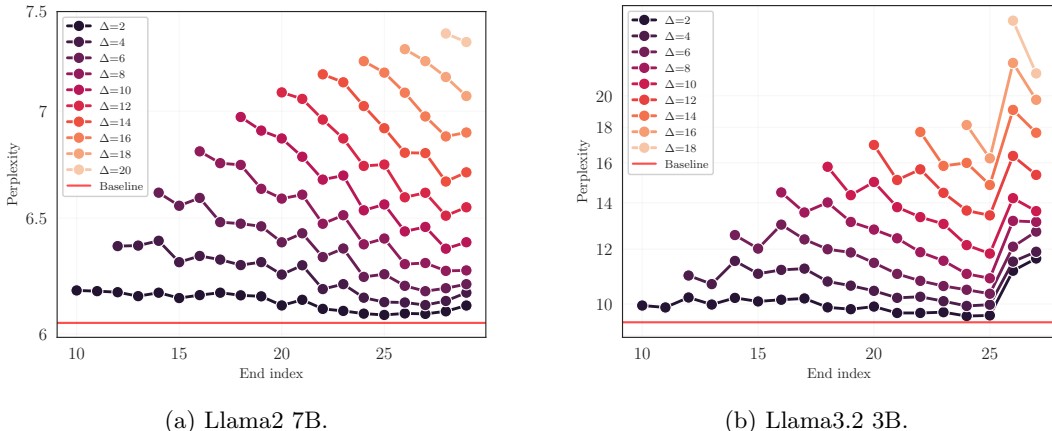

(a) Llama2 7B.

(b) Llama3.2 3B.

Figure 7: **Perplexity when running pairs of consecutive layers in parallel.** Perplexity of Llama2 7B and Llama3.2 3B models on the test set of RedPajama(Together Computer, 2023) when applying Layer Parallelism to $\Delta$ consecutive layers. The parallelized interval for each data point is [end index$-\Delta$, end index], where *end index* is the last layer in the LLM to which LP was applied.

Table 1: **5-shot In-Context Learning accuracies across standard benchmarks**. Effective Depth shows the minimum number of sequential operations from input to output after applying LP. We use the ablation on Fig 7 to choose the LP configurations that minimized the perplexity. *ifeval was evaluated on 0-shot performance.

| Eff. Depth | Speed | Avg | Rel. | MMLU | PiQA | Arc E. | Arc C. | WinoG | OBQA | hswag | GSM8K | ifeval* |
|---|---|---|---|---|---|---|---|---|---|---|---|---|
| **Llama 2 7B (Chat)** | | | | | | | | | | | | |
| 32 (base) | x1.00 | 53.80 | 1.00 | 47.27 0.4 | 77.69 1.0 | 79.63 0.8 | 49.06 1.5 | 72.06 1.3 | 33.00 2.1 | 58.87 0.5 | 22.97 1.2 | 43.65 |
| 27 (ours) | x1.15 | 53.89 | 1.00 | 47.46 0.4 | 77.09 1.0 | 78.03 0.9 | 48.81 1.5 | 71.82 1.3 | 33.80 2.1 | 58.39 0.5 | 22.21 1.1 | 47.36 |
| 26 (ours) | x1.19 | 53.25 | 0.99 | 47.24 0.4 | 76.66 1.0 | 77.48 0.9 | 47.01 1.5 | 71.51 1.3 | 34.00 2.1 | 57.79 0.5 | 19.03 1.1 | 48.56 |
| 25 (ours) | x1.23 | 52.33 | 0.97 | 47.67 0.4 | 76.33 1.0 | 77.19 0.9 | 46.50 1.5 | 70.09 1.3 | 33.20 2.1 | 57.24 0.5 | 14.78 1.0 | 47.96 |
| 24 (ours) | x1.28 | 49.62 | 0.92 | 45.47 0.4 | 76.55 1.0 | 75.67 0.9 | 43.69 1.5 | 67.88 1.3 | 30.40 2.1 | 55.82 0.5 | 9.63 0.8 | 41.49 |
| 23 (ours) | x1.31 | 47.71 | 0.89 | 42.71 0.4 | 75.24 1.0 | 74.28 0.9 | 41.13 1.4 | 66.54 1.3 | 30.80 2.1 | 54.06 0.5 | 6.97 0.7 | 37.65 |
| **Llama 3.2 3B (Instruct)** | | | | | | | | | | | | |
| 28 (base) | x1.00 | 60.88 | 1.00 | 59.56 0.4 | 77.09 1.0 | 79.29 0.8 | 46.50 1.5 | 70.24 1.3 | 30.80 2.1 | 52.53 0.5 | 64.75 1.3 | 67.15 |
| 24 (ours) | x1.12 | 57.16 | 0.94 | 59.14 0.4 | 76.22 1.0 | 76.60 0.9 | 44.88 1.5 | 70.09 1.3 | 30.0 2.1 | 51.31 0.5 | 45.87 1.4 | 60.31 |
| 23 (ours) | x1.15 | 55.44 | 0.91 | 59.05 0.4 | 75.46 1.0 | 76.18 0.9 | 44.71 1.5 | 68.98 1.3 | 27.80 2.0 | 51.16 0.5 | 35.71 1.3 | 59.95 |
| 22 (ours) | x1.19 | 51.59 | 0.85 | 56.12 0.4 | 75.30 1.0 | 75.04 0.9 | 45.73 1.5 | 66.77 1.3 | 28.80 2.0 | 50.88 0.5 | 10.01 0.8 | 55.64 |
| 21 (ours) | x1.23 | 48.03 | 0.79 | 50.72 0.4 | 74.70 1.0 | 72.56 0.9 | 40.19 1.4 | 64.09 1.4 | 28.40 2.0 | 49.47 0.5 | 3.11 0.5 | 49.04 |
| 20 (ours) | x1.28 | 45.76 | 0.75 | 40.70 0.4 | 74.10 1.0 | 70.75 0.9 | 39.42 1.4 | 62.75 1.4 | 29.00 2.0 | 47.55 0.5 | 3.11 0.5 | 44.48 |
| **Qwen3 4B (Instruct)** | | | | | | | | | | | | |
| 36 (base) | x1.00 | 63.72 | 1.00 | 70.16 0.4 | 76.44 1.0 | 84.76 0.8 | 58.79 1.5 | 66.30 1.3 | 37.20 2.0 | 52.77 0.5 | 84.99 1.0 | 42.09 |
| 31 (ours) | x1.13 | 57.72 | 0.91 | 68.87 0.4 | 74.59 1.0 | 81.82 0.8 | 53.67 1.5 | 65.98 1.3 | 35.20 2.2 | 50.10 0.5 | 53.75 1.0 | 35.49 |
| 30 (ours) | x1.15 | 55.45 | 0.87 | 67.49 0.4 | 74.97 1.0 | 81.84 0.9 | 52.39 1.4 | 65.11 1.3 | 33.40 2.1 | 48.95 0.5 | 36.77 0.9 | 38.13 |
| 29 (ours) | x1.18 | 51.84 | 0.81 | 63.56 0.4 | 74.70 1.0 | 79.88 0.9 | 50.43 1.4 | 63.22 1.4 | 33.00 2.2 | 47.25 0.5 | 17.97 0.8 | 36.57 |
| 28 (ours) | x1.21 | 49.09 | 0.77 | 53.95 0.4 | 74.32 1.0 | 79.29 0.9 | 48.98 1.6 | 62.75 1.3 | 30.0 2.1 | 45.24 0.5 | 12.66 0.5 | 34.65 |
| 27 (ours) | x1.25 | 44.68 | 0.70 | 44.09 0.4 | 72.96 1.0 | 76.68 0.9 | 44.54 1.5 | 59.67 1.4 | 29.60 2.0 | 43.22 0.5 | 3.56 0.5 | 27.82 |
| **Qwen3 14B (Instruct)** | | | | | | | | | | | | |
| 40 (base) | x1.00 | 68.75 | 1.00 | 78.83 0.4 | 80.90 1.0 | 87.75 0.8 | 66.21 1.4 | 74.51 1.3 | 40.40 2.1 | 61.33 0.5 | 82.26 1.2 | 46.52 |
| 35 (ours) | x1.12 | 65.04 | 0.95 | 77.92 0.4 | 79.82 1.0 | 86.95 0.9 | 64.68 1.5 | 73.56 1.3 | 38.80 2.0 | 58.25 0.5 | 61.64 1.1 | 43.76 |
| 34 (ours) | x1.15 | 66.29 | 0.96 | 77.42 0.4 | 78.89 1.0 | 85.61 0.9 | 62.71 1.5 | 73.88 1.3 | 37.80 2.0 | 57.92 0.5 | 75.36 1.0 | 47.00 |
| 33 (ours) | x1.18 | 64.64 | 0.94 | 75.88 0.4 | 79.05 1.0 | 85.61 0.9 | 60.58 1.4 | 72.69 1.4 | 37.40 2.0 | 56.93 0.5 | 69.83 1.0 | 43.76 |
| 32 (ours) | x1.21 | 62.84 | 0.91 | 73.70 0.4 | 78.89 1.0 | 84.81 0.9 | 59.73 1.4 | 71.03 1.4 | 36.80 2.0 | 55.75 0.5 | 64.67 0.9 | 40.17 |
| 31 (ours) | x1.24 | 60.11 | 0.87 | 71.56 0.4 | 78.56 1.0 | 85.52 0.9 | 59.98 1.5 | 68.59 1.3 | 36.40 2.0 | 54.12 0.5 | 48.22 0.8 | 38.01 |
| 30 (ours) | x1.27 | 55.18 | 0.80 | 65.43 0.4 | 77.58 1.0 | 83.84 0.9 | 55.89 1.5 | 67.32 1.3 | 34.60 2.1 | 51.97 0.5 | 23.43 0.8 | 36.57 |

Table 2: Benchmark accuracy restoration for LP-applied Qwen3 models. Fine-tuned entries use 4096 additional training steps; left columns report the fine-tuned accuracy, and right columns report the no-finetune/baseline values.

| Model | Eff. depth | Fine-tuned | | | No fine-tuning | | |
|---|---|---|---|---|---|---|---|
| | | MMLU | Arc C. | GSM-8K (%) | MMLU | Arc C. | GSM-8K (%) |
| Qwen3 4B | 36 (Baseline) | — | — | — | 70.16 0.4 | 58.79 1.5 | 84.99 1.0 |
| | 31 | 69.11 0.4 | 54.10 1.5 | 62.47 1.3 | 68.87 0.4 | 53.67 1.5 | 53.75 1.0 |
| | 30 | 68.43 0.4 | 54.10 1.5 | 56.56 1.4 | 67.49 0.4 | 50.43 1.4 | 36.77 0.9 |
| | 27 | 61.96 0.4 | 51.02 1.6 | 48.29 1.4 | 44.09 0.4 | 44.54 1.5 | 3.56 0.5 |
| Qwen3 14B | 40 (Baseline) | — | — | — | 78.83 0.4 | 66.21 1.4 | 82.26 1.2 |
| | 35 | 77.89 0.3 | 63.05 1.4 | 81.73 1.1 | 77.92 0.4 | 64.68 1.5 | 61.64 1.1 |
| | 32 | 74.38 0.4 | 59.30 1.4 | 71.27 1.3 | 73.70 0.3 | 59.73 1.4 | 64.67 0.9 |

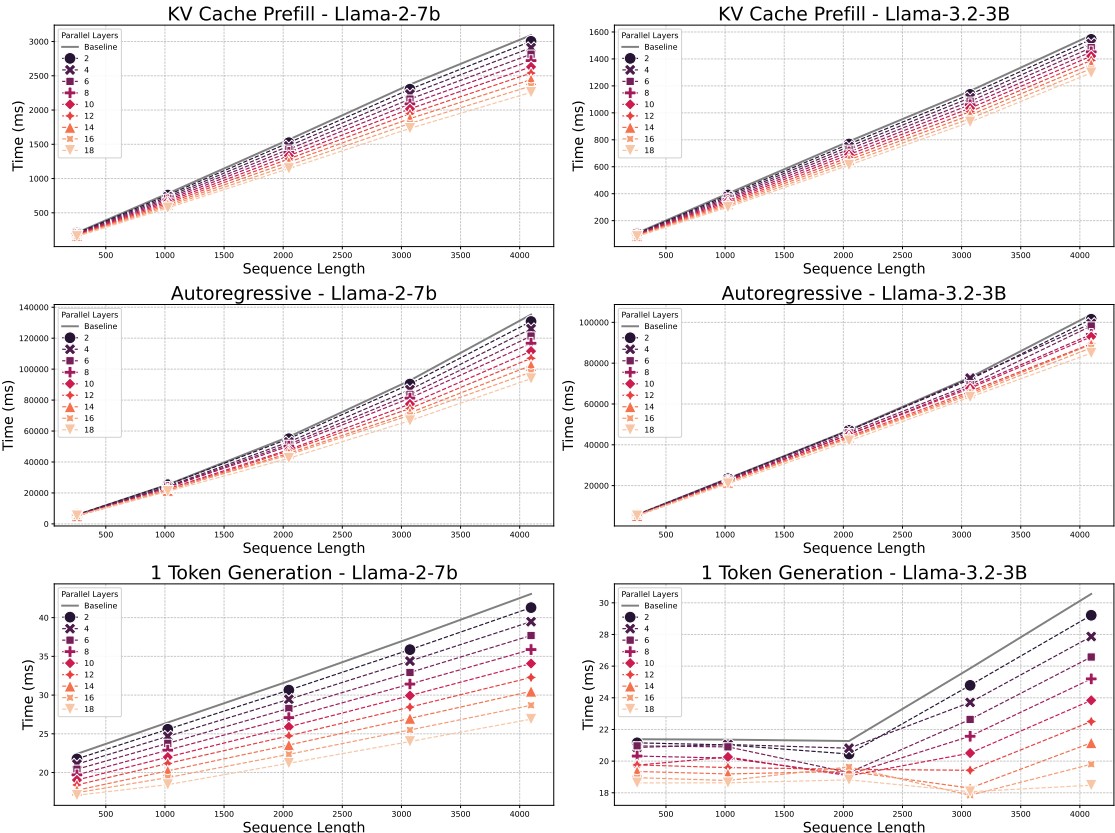

Figure 8: **Wall clock time to complete the following inference tasks**: KV Cache pre-filling, autoregressive generation, and single token generation with a pre-filled KV Cache. $\Delta$ indicates how many layers have been merged using LP (e.g. a $\Delta$ of 4 indicates that 2 groups of 2 layers have been converted to 2 effective layers). The gains in inference speed are roughly proportional to the amount of LP. The 1-token generation task for Llama 3.2 3B does not saturate the GPU compute until a sequence length of 2048. Even in this regime, LP benefits from considerable speed-ups.

## 5 Experiments & Results

In this section, we evaluate Layer Parallelism across three dimensions: inference speed improvements, impact on In-Context Learning performance, and the potential to recover model accuracy through targeted fine-tuning of parallelized layers.

**Experimental protocol.** For all our experiments, we use a node with x2 A100 SXM4 80Gb GPUs, x4 AMD EPYC 7742 CPUs, and 512Gb of RAM. We test for varying sequence lengths, up to 4096 (Llama's context window), with a batch size of 1 unless indicated otherwise. We consider two models of the Llama family: Llama2 7B, and Llama3.2 3B, as well as two sizes from Qwen3: 4B and 14B. Given a desired effective depth, we replace the required number of normal layers with LP layers. For Llama 2 7B and 3.2 3B, the LP layers are selected based on the configuration that minimized the PPL for a given amount of LP (Fig. 7). For Qwen3, LP is applied until the 4th to last decoder layer. The rest of the layers implement the tensor parallel approach as described in (Shoeybi et al., 2020). For evaluation, we measure the ICL 5-shot accuracies using the `lm-eval` package (Gao et al., 2024). We test the ICL accuracy of the models on several tasks: MMLU (Hendrycks et al., 2021), PiQA (Bisk et al., 2019), ARC Easy, ARC Challenge, Winogrande (Sakaguchi et al., 2021), OpenBookQA (Mihaylov et al., 2018), Hellaswag (Zellers et al., 2019), GSM-8K (Cobbe et al., 2021) and ifeval (Zhou et al., 2023). The perplexity of the models is always evaluated against a subset of the test set of RedPajama (Together Computer, 2023).

**Impact of LP on PPL and ICL accuracies.** We first examine how perplexity evolves when applying LP across layer sequences of different lengths and depths. Fig. 7 reveals a common optimal sequence end-index minimizing perplexity, found at layers 28 for Llama2 7B and 25 for Llama3.2 3B. Table 1 compares the In-Context Learning performance across models with varying effective depths. Performance declines gradually as LP increases, followed by a sharp drop beyond a certain threshold. Specifically, this occurs after reducing effective depth by 9 layers for Qwen3 14B, by 7 layers for Llama2 7B and Qwen3 4B, and by 5 layers for Llama3.2 3B. These results indicate that larger models are more robust to the computational graph modifications from LP, suggesting that our approach is likely applicable to current commercial-scale LLMs used in major deployments.

It is worth noting that, unlike the other benchmarks, GSM-8K already drops severely in performance when applying low amounts of LP. Recent mechanistic interpretability research shows that LLMs have special circuitry for math operations, localized in a small set of parameters (Stolfo et al., 2023; Yu & Ananiadou, 2024). (Christ et al., 2025) identify math specific parameters, and report a drop in accuracy of 17% when pruning them. We hypothesize that the changes in the computational graph by the use of LP in some of the late layers of the LLM interfere with these fragile and sparse subnetworks, while leaving general language competence largely intact.

**Impact on the inference speed.** We run an ablation over several configurations and input sequence lengths on Figure 8 to test the speed on three different tasks: KV-Cache pre-filling, autoregressive generation up to the sequence length(with KV-Cache) and 1-token generation with a pre-filled KV-Cache of the corresponding sequence length. Our ablations show that the speed gain is strongly correlated with the reduction of the effective depth of the model. For the effective depths of 25 ($\Delta = 14$) in Llama 2 7B, we observe an average speed-up of 1.29x at the largest sequence length in the 1-token generation task. Likewise, for an effective depth of 23 ($\Delta = 10$) in Llama 3.2 3B, we report a speed-up of 1.22x. For more aggressive parallelism, $\Delta = 18$ and $\Delta = 16$, we report a speed-up of 1.38x and 1.35x , at the expense of a large drop in ICL accuracy.

**Fine-tuning for performance recovery.** While LP provides speed improvements, associated architectural modifications may degrade model performance. To counteract this, we explored whether fine-tuning could effectively restore the original model's capabilities. We apply LP to some configurations of Qwen3-4B and Qwen3-14B (Table 2), and fine-tune the LP layers on randomly selected samples from the RedPajama training set (Together Computer, 2023). We employ a batch size of 32, a linear learning rate schedule starting at $1e-4$ and the AdamW optimizer (Loshchilov & Hutter, 2017). We observe a significant restoration of the benchmark accuracies for Qwen3-4B with an effective depth of 27, especially on GSM-8K, which recovered from near-zero levels. Less aggressive usage of LP results in a less pronounced recovery of the accuracy, and fails to fully recover the original model's performance. It is possible that additional fine-tuning, or smarter tuning strategies, could yield further improvements, but resource constraints limited the scope of our experiments.

## 6  Limitations

The effectiveness of our approach exhibits notable variations across model scales. Smaller models show reduced benefits, likely due to their less sparse activation patterns and more tightly coupled layer dependencies. This degradation becomes more pronounced as the LP sequence length increases, suggesting a practical upper limit to the number of layer pairs that can be effectively parallelized.

Regarding the fine-tuning, while some performance loss can be mitigated, we were unable to fully recover the baseline model's performance levels. This suggests fundamental trade-offs between computational efficiency and model capability that cannot be entirely eliminated through optimization, or that more involved fine-tuning strategies might be required.

Moreover, determining the 'true' effective depth—the optimal configuration of parallel layer pairs—remains an open challenge as there is no theoretical framework for predicting the optimal grouping strategy.

These limitations highlight important directions for future research, particularly in developing more robust methods for determining optimal layer groupings and investigating the interplay between our approach and other efficiency-oriented techniques.

## 7  Conclusion

In this work, we presented Layer Parallelism, a novel approach that exploits independence patterns between transformer layers to optimize LLM inference. By restructuring the computational graph to enable parallel execution of consecutive layer pairs through tensor parallelism, we achieved substantial speed improvements without model retraining. Our method reduced the effective depth of Llama 2 7B by 21% while maintaining 98% of the original performance (without fine-tuning), yielding up to a 1.29x improvement in inference speed for single-token generation with long sequences. Moreover, we show that we can recover some of the lost accuracy through naive fine-tuning.

These results challenge the conventional view that transformer layers must process information strictly sequentially, suggesting instead that certain layers can operate independently without significant performance loss. From a practical standpoint, LP offers a straightforward approach to improve inference efficiency in production environments. Future work could focus on developing theoretical frameworks to predict optimal layer groupings, investigating interactions with other efficiency techniques such as quantization, and understanding the fundamental principles behind layer independence. Despite its limitations, LP represents a practical advancement in making LLM deployment more efficient and economically viable.

## Acknowledgments

We would like to thank Benjamin Rio for the thoughtful discussions and help running some of the performance experiments. Ramón's research is supported by META.

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

# A  Theoretical Analysis of Layer Parallelism

This section provides a theoretical justification for Layer Parallelism (LP), analyzing the approximation error introduced by the LP computational graph and connecting it to the empirical observations in the main paper.

## A.1  Sequential vs. Layer-Parallel Computation

Consider two consecutive transformer decoder layers $\ell_k$ and $\ell_{k+1}$ in a pre-norm architecture. Let $A_k(\cdot)$ denote the attention residual and $F_k(\cdot)$ the feed-forward residual of layer $k$.

**Exact sequential computation.**  The standard forward pass computes:

$$u_k = x + A_k(x), \tag{2}$$
$$h_k = u_k + F_k(u_k), \tag{3}$$
$$u_{k+1} = h_k + A_{k+1}(h_k), \tag{4}$$
$$h_{k+1} = u_{k+1} + F_{k+1}(u_{k+1}), \tag{5}$$

where $h_{k+1} = T_{\text{seq}}(x)$ is the two-layer sequential output.

**Layer Parallelism computation.**  LP evaluates both attention modules at the shared input $x$, combines their outputs, and feeds the result to both FFN modules:

$$\tilde{u} = x + A_k(x) + A_{k+1}(x), \tag{6}$$
$$T_{\text{LP}}(x) = \tilde{u} + F_k(\tilde{u}) + F_{k+1}(\tilde{u}). \tag{7}$$

This matches the (LP) equation in §3 and the implementation in Fig. 2(b), where divergent paths share intermediate states.

## A.2  First-Order Error Analysis

The approximation error $\mathcal{E}(x) = T_{\text{seq}}(x) - T_{\text{LP}}(x)$ arises from evaluating submodules at different inputs. We decompose this error into three components.

**Component 1: Attention evaluation shift.**  In the sequential computation, $A_{k+1}$ is evaluated at $h_k = x + A_k(x) + F_k(u_k)$, whereas LP evaluates it at $x$. Defining $\Delta_1 = h_k - x = A_k(x) + F_k(u_k)$, a first-order Taylor expansion yields:

$$A_{k+1}(h_k) - A_{k+1}(x) \approx J_{A_{k+1}}(x)\,\Delta_1, \tag{8}$$

where $J_{A_{k+1}}(x)$ is the Jacobian of $A_{k+1}$ at $x$.

**Component 2: First FFN evaluation shift.**  In the sequential computation, $F_k$ is evaluated at $u_k = x + A_k(x)$, whereas LP evaluates it at $\tilde{u} = x + A_k(x) + A_{k+1}(x)$. Since $\tilde{u} - u_k = A_{k+1}(x)$:

$$F_k(u_k) - F_k(\tilde{u}) \approx -J_{F_k}(u_k)\,A_{k+1}(x). \tag{9}$$

**Component 3: Second FFN evaluation shift.**  In the sequential computation, $F_{k+1}$ is evaluated at $u_{k+1} = h_k + A_{k+1}(h_k)$, whereas LP evaluates it at $\tilde{u}$. The difference is:

$$u_{k+1} - \tilde{u} = F_k(u_k) + \big[A_{k+1}(h_k) - A_{k+1}(x)\big]. \tag{10}$$

Substituting the first-order approximation from (8):

$$F_{k+1}(u_{k+1}) - F_{k+1}(\tilde{u}) \approx J_{F_{k+1}}(\tilde{u})\big[F_k(u_k) + J_{A_{k+1}}(x)\,\Delta_1\big]. \tag{11}$$

**Total error bound.** Combining equations (8)–(11) and taking norms:

$$\|\mathcal{E}(x)\| \lesssim \underbrace{\|J_{A_{k+1}}\| \|\Delta_1\|}_{\text{attention shift}} + \underbrace{\|J_{F_k}\| \|A_{k+1}(x)\|}_{\text{FFN}_k \text{ shift}} + \underbrace{\|J_{F_{k+1}}\| \big(\|F_k(u_k)\| + \|J_{A_{k+1}}\| \|\Delta_1\|\big)}_{\text{FFN}_{k+1} \text{ shift}}, \qquad (12)$$

where Jacobian norms are operator norms evaluated at the appropriate inputs (suppressed for clarity).

### A.3 Implications for Layer Selection

The error bound (12) reveals when LP introduces minimal degradation:

**Small residual updates favor LP.** The bound depends on $\|\Delta_1\| = \|A_k(x) + F_k(u_k)\|$, $\|A_{k+1}(x)\|$, and $\|F_k(u_k)\|$. Layers where attention and FFN residuals are small relative to the residual stream contribute less error. This is consistent with the "residual stream dominance" phenomenon observed in deep transformers.

**Low Jacobian sensitivity favors LP.** The terms $\|J_{A_{k+1}}\|$ and $\|J_{F_k}\|$, $\|J_{F_{k+1}}\|$ measure how sensitive each submodule is to input perturbations. Layers that are relatively insensitive to their exact input introduce less LP error.

**Connection to CKA analysis (Fig. 5).** The CKA similarity between standard and counterfactual activations (with residual removed) serves as an empirical proxy for the Jacobian sensitivity. High CKA similarity indicates that the module output is relatively invariant to the upstream residual—precisely the condition under which the Jacobian terms in (12) are effectively small in the directions that matter. The plateau of high CKA values in mid-to-late layers (Fig. 5) corresponds to the region where LP is most effective (Fig. 4e).

**Error accumulation through the network.** Errors injected at layer $k$ propagate through all subsequent layers via the Jacobian chain:

$$\|\mathcal{E}_{\text{output}}\| \lesssim \|\mathcal{E}_k\| \prod_{\ell > k} \|I + J_{f_\ell}\|, \qquad (13)$$

where $f_\ell$ is the full residual map of layer $\ell$. This implies:

- **Avoid early layers**: Errors introduced early are amplified by many subsequent Jacobians.
- **Preserve a sequential tail**: The final layers before the output logits are typically most sensitive (the language modeling head amplifies perturbations), so leaving them sequential stabilizes the output distribution.

### A.4 Why Contiguous 2-Parallel Works Best

The "2-parallel" scheme (parallelizing consecutive pairs rather than arbitrary groups) succeeds because:

1. **Local ordering matters less than global ordering.** Shuffling experiments (Fig. 4a) show that adjacent layers are more interchangeable than distant ones, likely because they operate at similar levels of abstraction.
2. **Error terms remain bounded.** Within a single LP pair, the error is first-order in residual magnitudes. Chaining $n$ LP pairs gives $n$ independent first-order errors rather than a single large error from parallelizing all $2n$ layers simultaneously.
3. **Intermediate synchronization corrects drift.** Between LP pairs, the outputs are summed and re-normalized, preventing error accumulation within the LP region.

This explains why parallelizing triplets performs worse (as noted in §3): the second-order cross-terms become significant, and there is no intermediate synchronization to correct the trajectory.

### A.5 Connection to GSM-8K Degradation

The disproportionate drop in GSM-8K accuracy under LP (Table 1) is consistent with recent findings that mathematical reasoning relies on sparse, localized circuits (Stolfo et al., 2023; Christ et al., 2025). These circuits likely have:

- Larger effective Jacobians in the relevant directions (high sensitivity to precise intermediate states).
- Less redundancy, so the LP approximation error is not absorbed by parallel pathways.

General language competence, by contrast, is distributed across many redundant pathways and is therefore more robust to the input perturbations introduced by LP.

# B    Comparison with Other Tensor Parallelism Optimizations

In this section, we compare Layer Parallelism (LP) with recent methods that aim to reduce communication overhead in tensor-parallel LLM inference: Sync-Point Drop (SPD) (Kim et al., 2025), selective low-bit communication (Dong et al., 2024), and microscaling (MX) format compression (Hansen-Palmus et al., 2024).

## B.1    Method Overview

**Sync-Point Drop (SPD).**   Kim et al. (2025) selectively removes the all-reduce synchronization after the self-attention output projection, retaining only the FFN synchronization per block. They introduce modified block designs to minimize information loss and classify blocks into three sensitivity categories (in-sensitive, sensitive, extremely sensitive), applying block-to-block distillation to recover accuracy in sensitive layers.

**Selective Low-bit Communication.**   Dong et al. (2024) compress communicated activations by quantizing most features to INT4 while keeping a small fraction (1/64) of high-range outlier features in BF16. This reduces communication from 16 bits to ∼4.2 bits per value on average, preserving outlier information critical for model performance.

**MX Format Compression.**   Hansen-Palmus et al. (2024) apply microscaling (MX) quantization formats (FP4/FP5 with block-wise scaling) to compress activations before inter-device communication, achieving 3.5–4.5× compression ratios.

**Layer Parallelism (Ours).**   LP restructures the computational graph to execute consecutive layer pairs in parallel, reducing the number of sequential synchronization points. Unlike quantization-based methods, LP modifies the *computation order* rather than the *communication encoding*.

## B.2    Quantitative Comparison

Table 3 compares results across methods. Direct comparison is challenging due to different models, hardware configurations, and evaluation metrics; we match the closest available configurations.

Table 3: Comparison of tensor parallelism optimization methods. Speedup is relative to standard tensor parallelism. For accuracy: SPD and LP report average zero-shot accuracy change; Low-bit reports performance retention; MX reports perplexity increase. Results marked [†] are at 70% SPD application; [‡] indicates hardware-dependent results on PCIe-connected GPUs (L4); [§] indicates NVLink-connected GPUs (A100).

| Method | Model | GPUs | Speedup | Accuracy Impact | Mechanism |
|---|---|---|---|---|---|
| *7B-scale Models* | | | | | |
| SPD[†] | LLaMA2-7B | 8 | 1.10× | −1.0% avg acc | Drop attn sync |
| Low-bit | LLaMA2-13B | 8 | — | 99.5% retained | Quant. comm |
| MX Compress | LLaMA2-7B | 2[‡] | 1.03× | +3.2% PPL | Quant. comm |
| **LP (Ours)** | LLaMA3.2-3B | 2 | **1.19×** | −2.5% avg acc | Parallel layers |
| *13B-scale Models* | | | | | |
| SPD[†] | LLaMA2-13B | 8 | 1.12× | −1.0% avg acc | Drop attn sync |
| Low-bit | LLaMA2-13B | 8 | — | 99.5% retained | Quant. comm |
| MX Compress | LLaMA2-13B | 4[‡] | 2.05× | +3.2% PPL | Quant. comm |
| **LP (Ours)** | Qwen3-14B | 2 | **1.15×** | −4.0% avg acc | Parallel layers |
| *70B-scale Models* | | | | | |
| SPD[†] | LLaMA2-70B | 8 | 1.20× | −0.9% avg acc | Drop attn sync |
| MX Compress | LLaMA2-70B | 8[‡] | 1.83–2.08× | +1.7% PPL | Quant. comm |
| MX Compress | LLaMA2-70B | 4[§] | 0.56–0.70× | +1.7% PPL | Quant. comm |

Table 4 provides a detailed accuracy comparison on common benchmarks where available.

Table 4: Accuracy comparison on zero-shot benchmarks. Values show absolute accuracy (%) or relative change from baseline. SPD results use ZS+B2B configuration at 70% SPD; Low-bit uses INT4+Selected BF16 at 4.2 bits.

| Method | Model | ARC-e | ARC-c | HellaSwag | WinoGrande | Avg |
|--------|-------|-------|-------|-----------|------------|-----|
| *Baseline (no optimization)* | | | | | | |
| — | LLaMA2-13B | 79.5 | 48.7 | 60.0 | 72.2 | 65.1 |
| — | Gemma 2 27B | 87.7 | 62.4 | 65.4 | 79.1 | 73.7 |
| *Low-bit Communication (Dong et al., 2024)* | | | | | | |
| INT4+Sel. BF16 | LLaMA2-13B | 79.1 | 47.4 | 59.5 | 72.9 | 64.7 |
| | | (−0.5%) | (−2.8%) | (−0.9%) | (+1.0%) | (−0.6%) |
| INT4+Sel. BF16 | Gemma 2 27B | 86.5 | 61.0 | 63.9 | 76.5 | 72.0 |
| | | (−1.4%) | (−2.2%) | (−2.3%) | (−3.3%) | (−2.3%) |
| *SPD (Kim et al., 2025) at 70% blocks* | | | | | | |
| ZS+B2B | LLaMA2-13B (8-GPU) | | | — | | ∼65% |
| | | | | | | (<1% drop) |
| ZS+B2B | LLaMA2-70B (8-GPU) | | | — | | ∼66% |
| | | | | | | (−0.9%) |

## B.3 Key Observations

**Complementary approaches.** The methods target different bottlenecks: LP and SPD reduce the *number* of synchronization points, while low-bit and MX compression reduce the *size* of each synchronization. These are largely orthogonal and could be combined—for instance, applying LP to reduce sync-points by 50%, then using INT4 quantization to compress remaining communications by 4×.

**Hardware sensitivity.** Communication compression methods show strong hardware dependence. Hansen-Palmus et al. (2024) report 2× TTFT speedup on PCIe-connected L4 GPUs (64 GB/s bandwidth) but *slowdown* on NVLink-connected A100s (600 GB/s bandwidth) due to quantization overhead exceeding communication savings. In contrast, LP and SPD benefit from reduced sync-points regardless of interconnect speed, though gains are more pronounced on slower interconnects.

**Model size scaling.** Both SPD and LP show improved robustness on larger models:

- **SPD**: 44% of blocks are in-sensitive in LLaMA2-7B vs. 75% in LLaMA2-70B (Kim et al., 2025)
- **LP**: Larger models (Qwen3-14B) tolerate greater effective depth reduction than smaller models (LLaMA3.2-3B) before sharp accuracy drops (see Table 1)

**Trade-off characteristics.** Each method exhibits distinct accuracy–speed trade-offs:

- **SPD**: Graceful degradation; requires per-block sensitivity analysis and optional distillation
- **Low-bit**: Minimal degradation (∼0.5–2%) by preserving outlier features; no speedup measured
- **MX Compress**: Perplexity increases 1–3%; speedup highly hardware-dependent
- **LP (Ours)**: Uniform accuracy drop across layers; GSM-8K disproportionately affected; fine-tuning recovers ∼50% of loss

## B.4 Combining LP with Communication Compression

A promising direction is combining LP with quantization-based compression. Table 5 estimates potential combined benefits.

We leave empirical validation of combined LP + quantization to future work.

Table 5: Estimated combined speedup from LP + communication quantization on bandwidth-constrained hardware. LP reduces sync-points; INT4/MX4 reduces per-sync data volume by ∼4×.

| Optimization | Sync Reduction | Comm. Compression | Est. Speedup |
|---|---|---|---|
| Baseline (TP) | 0% | 1× | 1.0× |
| LP only | 25–50% | 1× | 1.15–1.25× |
| INT4/MX4 only | 0% | 4× | 1.5–2.0× |
| **LP + INT4/MX4** | 25–50% | 4× | **1.8–2.5×** |

## C  Ablation: Tokens per second

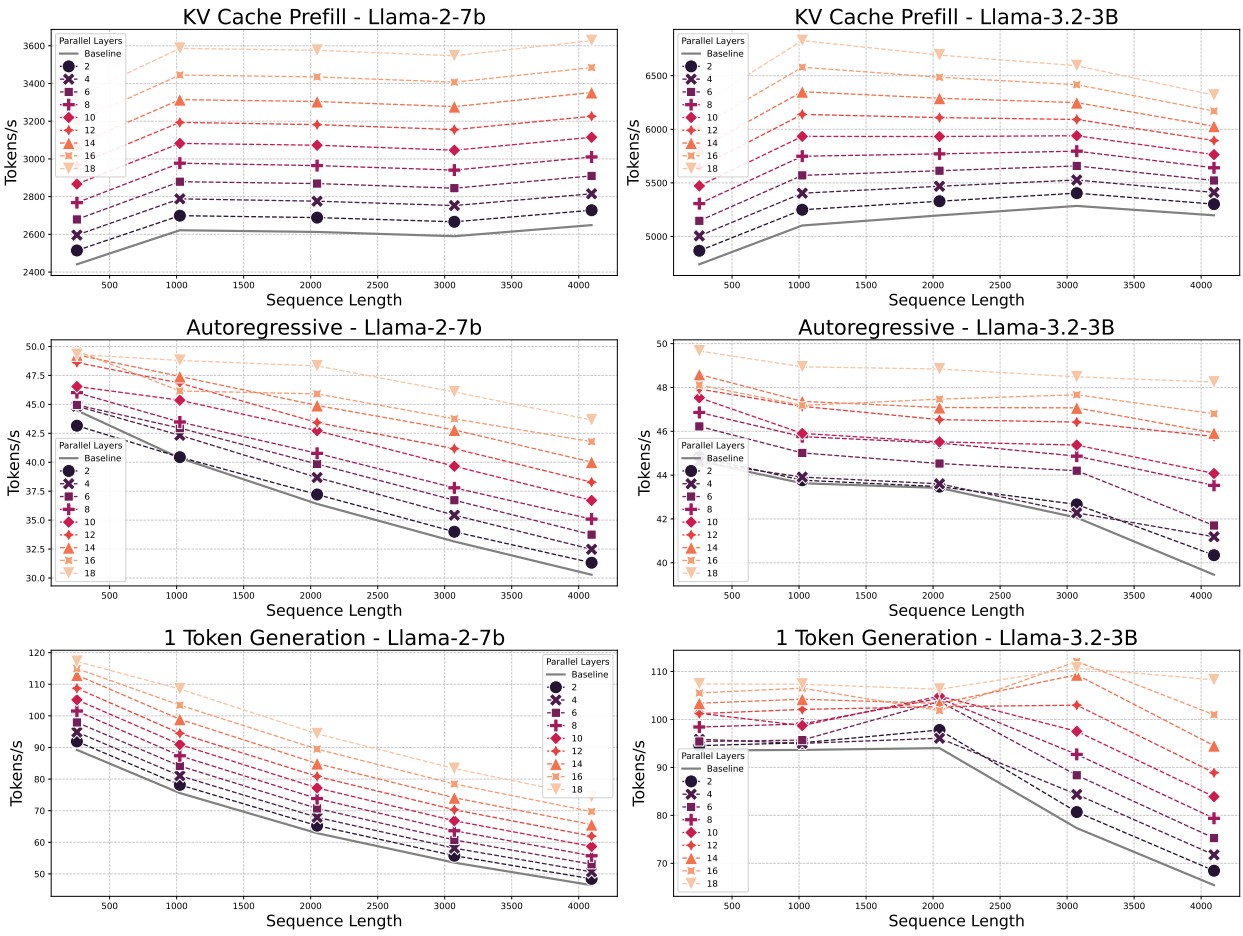

Figure 9: Tokens per second when completing the following inference tasks: KV Cache pre-filling for a given sequence length, autoregressive generation up to the indicated sequence length, and single token generation with a pre-filled KV Cache of the indicated sequence length. The baseline is the original model with all layers making use of Tensor Parallelism. The Parallel Layers number (Δ) indicates how many layers have been merged using Layer Parallelism (e.g. a Δ of 4 indicates that 2 groups of 2 layers have been converted to 2 effective layers). The number of tokens is computed as the sum of the input tokens and the output tokens for each forward pass.

# D Generalization to multiple GPUs

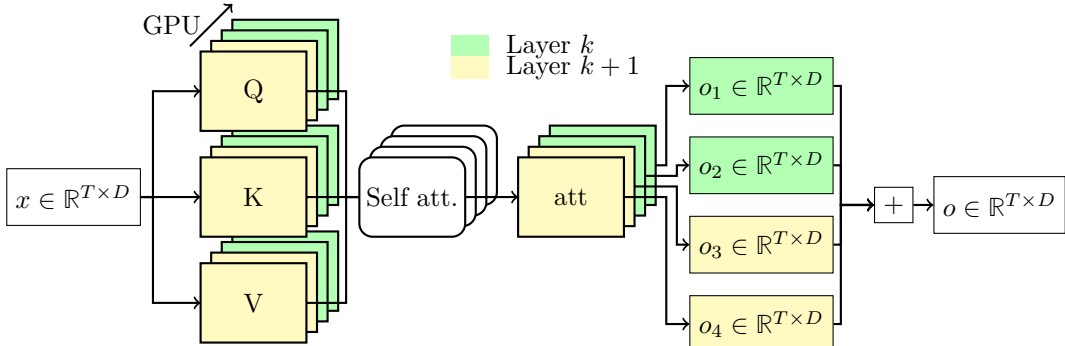

Figure 10: Layer Parallelism in the case of parallelizing two layers over four accelerators. The stacked layers represent the tensor parallelism, and the colors indicate the processing of different previously contiguous layers. $Q, K, V, \text{att} \in \mathbb{R}^{T \times \frac{2D}{g}}$, where $D$ is the feature dimension and $g$ is the total number of accelerators.

Layer Parallelism allows one to allocate $N \geq 1$ accelerators for each layer. The implementation remains the same, but now each layer is parallelized using tensor parallelism over its assigned accelerators. Note that both reduction operations (tensor parallel and layer parallel) are nicely executed with a single all-reduce call.

To confirm that the proposed scheme scales to commodity 4-GPU servers, we benchmark Llama 2 7B on a node with $4\times$ NVIDIA A100 80 GB PCIe accelerators while running the 4-GPU LP implementation. For each configuration we measure the wall-clock time of the decoding workload, normalize it to the vanilla $\Delta = 32$ setting, and report the resulting relative throughput in Table 6. The gains steadily increase as we parallelize more layers (e.g., $\Delta = 23$ reaches 1.46×), illustrating the benefit of halving the number of synchronization steps per block; however, these aggressive settings correspond to the larger accuracy drops discussed in the main text, so moderate $\Delta$ values offer a better accuracy–speed trade-off.

Table 6: Relative throughput of 4-GPU Layer Parallelism on Llama 2 7B measured on $4\times$ A100 80 GB PCIe GPUs. $\Delta$ denotes the number of layers replaced by LP pairs, with $\Delta = 0$ serving as the baseline ($x1.00$).

| $\Delta$ | Rel. Speed |
|---|---|
| 0 | x1.00 |
| 10 | x1.24 |
| 14 | x1.34 |
| 18 | x1.46 |

# E   Acceleration source

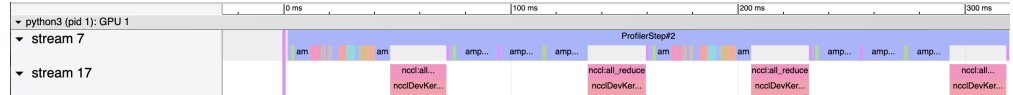

(a) Flame chart of running two standard Tensor-Parallel LLama decoder layers.

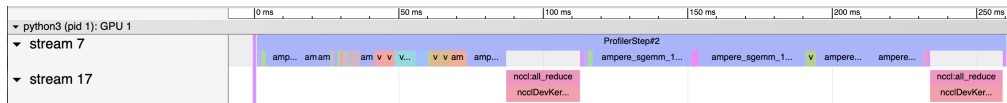

(b) Flame chart of running two Llama 3 decoder layers with our Layer Parallelism approach.

Figure 11: Comparison of Flame Graphs when running two consecutive Llama 3.2 3B decoder layers with vanilla tensor parallelism (Fig. 11a), and our Layer Parallelism approach (Fig. 11b). Note that the time axis scale is different between both graphs. These results were obtained on a workstation using x2 RTX 4090s.

Figure 11 illustrates flame graphs comparing two consecutive Llama 3.2 3B decoder layers using vanilla tensor parallelism and our LP approach. The profiling data summarized in Table 7 reveals that the primary source of acceleration in our LP method stems from reducing the total number of `all-reduce` synchronization operations across GPUs. Specifically, the vanilla tensor parallel approach performs synchronization at every decoder layer, resulting in higher cumulative synchronization overhead due to the fixed latency costs. Size-independent latency is dominated by GPU kernel-launch overhead, per-hop interconnect round-trip latency and host/stream synchronization imposed by the calling context. In contrast, our Layer Parallelism implementation runs pairs of layers simultaneously, effectively halving the number of synchronization points. This reduction in synchronization leads to a significant drop in synchronization time from 100.8ms to 50.7ms, directly contributing to the observed improvement in inference speed.

Additionally, Layer Parallelism enables fusion of certain computation kernels—particularly attention and MLP operations across parallelized layers—which further marginally reduces the computation time from 217ms to 208.7ms. Although these computational gains are modest compared to the savings achieved through fewer synchronization operations, kernel fusion further optimizes hardware utilization and enhances overall throughput.

Table 7: Profiling results comparing vanilla Tensor Parallel and Layer Parallel implementations on two consecutive Llama 3.2 3B decoder layers.

| Approach | Total Time (ms) | Sync Time (ms) | Computation Time (ms) |
|---|---|---|---|
| Tensor Parallel | 317.8 | 100.8 | 217.0 |
| Layer Parallel (Ours) | 259.4 (x1.23) | 50.7 (x1.99) | 208.7 (x1.04) |

