# OpenReview forum: "Leveraging the True Depth of LLMs"
_TMLR — Accepted by TMLR_

### Review · Reviewer_kBxv · 2025-10-19

**Summary Of Contributions:**

The paper explores how the computational graph of trained LLMs can be modified to run inference more efficiently on two GPUs.

The authors discovered that two consecutive transformer layers can be run in parallel (summing their activations), and running multiple pairs of layers in parallel impacts model quality only to a small degree, while decreasing inference time by reducing communication between the GPUs.

Llama and Qwen models ranging from 3B to 14B are investigated. In some cases with more pronounced performance degradation, fine-tuning part of the network can recover part of the difference.

**Additional Comments:**

1. Section 4 mentions "LP Multi-Head Attention", but the paragraph text and Figure 5 do not account for multiple heads.

**Audience:**

Yes

**Audience Explanation:**

It is an interesting observation that transformer layers of a trained LLM can be run in parallel, even though these networks have been trained with layers applied strictly sequentially. These networks are widely used, and new approaches to reduce the inference compute load have a potential for a large impact.

**Broader Impact Concerns:**

I don't see any ethical implications specific to layer parallelism that would require adding a Broader Impact Statement to the paper.

**Claims And Evidence:**

No

**Claims Explanation:**

I think that the broad claim "This approach, requiring no retraining, boosts inference throughput by 1.05x–1.20x while maintaining 95-99% of the original model’s accuracy on standard benchmarks." requires more evidence than what is provided in the paper. The paper only examines two models at a limited scale on few benchmarks, and inference speedup is only demonstrated in a single setup.

**Requested Changes:**

1. For better support of the central claim of the abstract, I would like to see the study extended in (some of) the following ways:

   a. Better display of the existing data – this could be improved by adding (or extending) a table to show the following numbers in a single place:  relative reduction of effective depth, relative reduction of metrics (with some estimate of variance), and relative speedup.

    b. How were the benchmarks from Table 1 chosen (the benchmarks partially overlap with the ones from [Llama 2](https://arxiv.org/abs/2307.09288), with slightly different numbers, but the benchmarks reported in [Qwen 3](https://arxiv.org/abs/2505.09388) are quite different)? Why are GSM8K and ifeval not included for the first two models?

    c. Hoes does layer parallelism perform with 1 GPU? How does it perform with 4 GPUs?

    d. How does layer parallelism perform with larger models? Does it work with MOEs?

    e. By how much does fine-tuning improve the results with models other than Qwen3 4B?

2. Appendix Section C states "that the primary source of acceleration in our LP method stems from reducing the total number of all-reduce synchronization operations across GPUs". Since this is of central importance for the approach, I would like to see more detailed explanations as on the exact mechanism of this acceleration. It seems that running layers in parallel reduces the number of synchronization points, but increases the data transferred per synchronization by the same factor.

3. For better readability, I would propose to add a third formula after (2, PAR) that defines the exact computation that was used for the reported results. Figure 2b should reference that formula (and maybe even moved out of the  "Related work" section).

---

> ### Author Response · Authors · 2025-11-16
> **Author response to Reviewer kBxv**
>
> We thank the reviewer for their thoughtful and constructive feedback. We have revised the manuscript accordingly and address each point in detail below.
>
> ## Abstract and central claim
>
> We have updated the abstract to avoid an overly broad claim and to align it more precisely with the evidence presented. The relevant sentence now reads:
>
> > “This approach, requiring no retraining, yields a 1.19× throughput gain on Llama 2 7B while only reducing by 1.5% the average benchmark accuracy.”
>
> This revised statement is directly supported by the revised Table 1 and the corresponding speed measurements.
>
> ## Requested Changes
>
> > a. Better display of the existing data …
>
> Table 1 and now reports the standard deviation for each benchmark under 5-shot evaluation setting. We also added a new column that explicitly reports the speed-up for each LP configuration.
>
> > b. Benchmark selection and evaluation
>
> They were chosen to cover a mix of knowledge, reasoning, and commonsense tasks and allow for direct comparison to previously reported results.
>
> We use Hugging Face’s `transformers` library together with the EleutherAI `lm-evaluation-harness` for all evaluations. We suspect the discrepancies with the Qwen3-reported numbers arise from differences in the inference stack between the original Qwen3 release and the Hugging Face implementation. In our work, we are primarily interested in the **relative** performance drop between the base model and its LP-transformed counterpart, and we therefore keep the evaluation framework fully consistent across all configurations.
>
> In earlier iterations we evaluated base models, which lack chat capabilities, and omitted GSM8K for those models. We now focus on the more practically relevant instruction/chat variants and have removed the base models from Table 1. The updated table reports results only for Llama 2 7B (Chat), Llama 3.2 3B (Instruct), Qwen3 4B, and Qwen3 14B.
>
> > c. How does layer parallelism perform with 1 GPU? How does it perform with 4 GPUs?
>
> LP is designed specifically for the tensor-parallel setting. It cannot be applied in the single-GPU case.
>
> For the 4-GPU setting, we added an ablation in Appendix B reporting the speed-ups for Llama 2 7B on a node with 4× A100 80 GB PCIe GPUs. The table is:
>
> | Δ | Rel. Speed |
> | - | - |
> | 32 | ×1.00 |
> | 27 | ×1.24 |
> | 25 | ×1.34 |
> | 23 | ×1.46 |
>
> > d. Bigger models and MoEs
>
> In this work, we evaluate models up to 14B parameters (Qwen3 14B). Due to resource constraints, we were not able to extend our experiments to larger models or to current open-source MoE architectures. We therefore cannot make empirical claims about LP’s behavior in those regimes and explicitly acknowledge this as a limitation.
>
> > e. Fine-tuning
>
> We extended our fine-tuning study to cover additional LP configurations of Qwen3 4B (Δ = 31, 30, 27) and Qwen3 14B (Δ = 35, 32). (some omitted due to text limit). We have updated the corresponding paragraph and now include the following table:
>
> | Model     | Eff. depth    | FT MMLU | FT Arc C. | FT GSM-8K (%) | No-FT MMLU | No-FT Arc C. | No-FT GSM-8K (%) |
> | --------- | ------------- | ------- | --------- | ------------- | ---------- | ------------ | ---------------- |
> | Qwen3 4B  | 31            | 69.11   | 54.10     | 62.47         | 68.87      | 53.67        | 53.75            |
> | Qwen3 4B  | 27            | 61.96   | 51.02     | 48.29         | 44.09      | 44.54        | 3.56             |
> | Qwen3 14B | 35            | 77.89   | 63.05     | 81.73         | 77.92      | 64.68        | 61.64            |
>
> All fine-tuning runs use 4,096 additional training steps. Overall, we observe that naive fine-tuning with this modest budget primarily recovers performance on GSM8K (math) while providing only partial recovery on the broader benchmarks. We highlight this explicitly in the revised text.
>
> > 2. Source of acceleration
>
> Collective operations such as al-reduce incur a non-negligible fixed latency cost per call (synchronization, kernel launch, ring-setup, ...). LP effectively reduces this fixed latency cost by half. We have expanded Appendix C to reflect this.
>
> > 3. Equation readability
>
> We have incorporated the explicit LP equation immediately after the PAR equation and now reference it in Figure 2b:
>
> $$
> \hat{\mathbf{y}} = \mathbf{x} + A_k(\mathbf{x}) + F_k(\mathbf{x} + A_k(\mathbf{x})) + A_{k+1}(\mathbf{x}) + F_{k+1}(\mathbf{x} + A_{k+1}(\mathbf{x})) \quad \mathrm{(PAR)}
> $$
>
> $$
> \hat{\mathbf{y}} \approx \mathbf{x} + A_k(\mathbf{x}) + A_{k+1}(\mathbf{x}) + F_k(\mathbf{x} + A_k(\mathbf{x}) + A_{k+1}(\mathbf{x})) + F_{k+1}(\mathbf{x} + A_k(\mathbf{x}) + A_{k+1}(\mathbf{x})) \quad \mathrm{(LP)}
> $$
>
> > Section 4 mentions “LP Multi-Head Attention”, but the paragraph text and Figure 5 do not account for multiple heads.
>
> Section 4 builds on Megatron-TP, which distributes the heads across GPUs. Figure 5 does not show the heads to avoid clutter. We added a clarifying sentence mentioning that $n_h$ must be divisible by the amount of GPUs.

---

### Review · Reviewer_3x6E · 2025-10-27

**Summary Of Contributions:**

The paper proposes a new technique for speeding up LLM inference by exploiting hidden redundancy in transformer architectures, which they call Layer Parallelism (LP). The core idea is that several adjacent transformer layers behave independently so the model can run pairs of consecutive layers in parallel instead of processing them strictly one after another, without retraining. By reorganizing the computational graph to enable this parallel execution of consecutive layer pairs through tensor parallelism, the authors achieve significant inference speedup with minimal performance degradation.

Strengths:
- LP introduces a new dimension of parallelism - across consecutive layers. This had not been explored in previous works.
- This method works on existing models trained on the transformers architecture, without retraining, and leverages tensor parallelism
- It opens up a new area for exploration - how tensor parallelism in transformers can be further optimized, especially in distributed inference

Weaknesses:
- The paper lacks a theoretical justification for why certain layers can be safely grouped, although it is supported empirically
- Benchmark selection is broad but could also include code generation, long-context, multi-turn dialogue tasks to truly test sequential depth
- The authors test LP on decoder-only architectures, so it’s unclear whether similar redundancy patterns exist in encoder-decoder or multimodal architectures

**Audience:**

Yes

**Audience Explanation:**

Yes. The paper is likely to be of interest to the TMLR audience, especially those interested in LLM efficiency and architecture optimization.

**Claims And Evidence:**

Yes

**Claims Explanation:**

Mostly, yes. Empirical claims about speedups and the existence of a usable accuracy/speed tradeoff are well-supported. The paper provides convincing wall-clock and benchmark evidence that this method can speed inference on real models without retraining, and that light fine-tuning helps recover quality. However, the broader claims about general applicability, why the effect occurs, and how to choose layers automatically are not yet fully supported.

**Requested Changes:**

Major -
1. Currently, the assumption of partial independence between layers is only supported empirically. Adding a lightweight analysis, such as activation correlation between layers, or attention pattern similarity could strengthen the paper in my view.

Minor -
1. For reproducibility, it would be good to report mean and standard deviation across multiple runs rather than single-run metrics reported in the paper
2. In addition to the wall-clock time, it would be good to see energy usage, GPU utilization, and/or memory usage to understand cost saving better

---

> ### Author Response · Authors · 2025-11-16
> **Author response to Reviewer 3x6E**
>
> We thank the reviewer for their thoughtful and constructive feedback. In particular, we are grateful for the suggestion to analyze activation similarity using CKA; incorporating this additional experiment has strengthened the empirical support for our LP hypothesis.
>
> ## Requested changes
>
> ### Analysis on activation correlation
>
> We applied CKA on the activations of Qwen3-4B to further support the hypothesis that the MHA and FFN modules exhibit a degree of independence with respect to the residuals of the preceding modules. Further experimental details and results are provided in our reply to reviewer vws1.
>
> ### Report mean and standard deviation
>
> We re-ran all the experiments in Table 1 and now report the standard deviation for each benchmark under the 5-shot evaluation setting.
>
> ### Energy usage, GPU utilization, and memory usage
>
> * **Energy usage and GPU utilization.** These metrics are tightly coupled to the time spent executing GPU kernels. Figure 11 and Table 3 in Appendix C show the time spent running kernels and the time GPUs remain idle due to all-reduce synchronization. While kernels are executing (for sufficiently large batch sizes), GPU compute is saturated, and energy consumption is close to the maximum rated level.
>
> * **Memory usage.** During inference, only the model weights and the current layer’s activations need to reside in memory. Under LP, the weight memory remains unchanged relative to TP: each GPU stores half as many layers but with twice as many weights per layer. However, the activation size for each LP layer is doubled, which increases the peak memory footprint as soon as a single LP block is used. Because these activations are only resident while the current layer is being computed and are discarded immediately afterward, the peak activation memory does not grow with the number of LP layers. We therefore believe a more detailed study of memory usage is not critical for understanding LP’s practical impact.

---

### Review · Reviewer_vws1 · 2025-11-03

**Summary Of Contributions:**

This paper proposes Layer Parallelism (LP), a method to accelerate large language model inference by exploiting the weak dependency between consecutive Transformer layers. Below are my reviews of this paper:

### **Strengths**

1. **Well-motivated and novel idea.**
   The paper provides an interesting perspective: when shuffling the middle layers of a Transformer, the model’s perplexity only increases slightly (by 2.9), while other operations such as pruning, merging, or removing layers lead to significant performance degradation.
   This observation supports the existence of **layer decoupling**, inspiring the authors to propose **parallel computation across adjacent Transformer layers** to improve efficiency.
   I like this idea—it is simple, intuitive, and can be directly applied to existing large models **without retraining**.

2. **Clear writing and organization.**
   The paper is well written and easy to follow. The *Limitations* section clearly acknowledges the constraints and open problems of the proposed method.

3. **Comprehensive experiments.**
   The authors evaluate the method on multiple open LLM families (Llama2, Llama3.2, Qwen3), examining both **perplexity trends** and **In-Context Learning (ICL)** performance.


### **Weaknesses**

1. **Lack of theoretical foundation for “Layer Decoupling” and parallel computation.**
   Although the empirical results suggest that intermediate layers are loosely coupled, the paper offers **no theoretical explanation** for this phenomenon.
   In addition, the transition from Equation (1) to Equation (2) is an **approximation**—although it empirically works but without theoretical grounding (e.g., representation similarity analysis, information flow metrics), the claim remains observational rather than principled, limiting its generalizability.

2. **Some experimental results are not very convincing.**
   For example, in Table 2, Qwen3-4B with an effective depth of 27 shows a **large performance drop** (from 70 to 44 average accuracy).
   Even after 8 K fine-tuning steps, the model only recovers to 61.34, still far from the baseline.
   It would be important to discuss **whether the efficiency gain justifies such degradation**—i.e., is the trade-off between speed-up and accuracy worthwhile?

3. **Lack of comparison with other tensor-parallel or efficiency methods.**
   The paper does not compare LP with existing tensor-parallel optimizations [1][2][3].
   In the *Impact on Inference Speed* section, LP achieves 1.29× (Llama 2 7B), 1.22× (Llama 3.2 3B), and up to 1.35–1.38× speed-up for larger Δ, but it is unclear how much accuracy is lost for each setting.
   Since LP inevitably introduces performance degradation, a **direct trade-off comparison** with prior efficiency-improvement methods would strengthen the paper’s contribution.

[1] Hansen-Palmus, Jan, et al. "Communication Compression for Tensor Parallel LLM Inference." arXiv preprint arXiv:2411.09510 (2024).
[2] Kim, Han-Byul, et al. "SPD: Sync-Point Drop for Efficient Tensor Parallelism of Large Language Models." arXiv preprint arXiv:2502.20727 (2025).
[3] Dong, Harry, et al. "Towards low-bit communication for tensor parallel llm inference." arXiv preprint arXiv:2411.07942 (2024).

**Audience:**

Yes

**Audience Explanation:**

This is a paper about improving the inference efficiency of large language models.

**Claims And Evidence:**

No

**Claims Explanation:**

Although the authors claim that layer parallelism improves inference speed, the trade-off between the speed gain and the degradation of model performance remains unclear. The experimental section does not provide a detailed correspondence between inference speed improvements and the extent of performance loss.

**Requested Changes:**

Provide theoretical justification for the PAR approximation.
The transition from Equation (1) to Equation (2) is currently supported only by empirical observations. The authors could strengthen their argument by analyzing layer similarity (e.g., CKA/SVCCA), Jacobian alignment, or mutual information between adjacent layers to demonstrate that their outputs are indeed loosely coupled. A theoretical or analytical framework would make the layer-parallel assumption more principled and generalizable.

Quantify the trade-off between inference speed and accuracy degradation.
While the paper reports overall speed-up ratios (e.g., 1.29×, 1.22×, 1.35×), it does not explicitly link these gains to the corresponding performance drops. Presenting a clear speed–accuracy curve or table showing how inference efficiency scales with accuracy loss would greatly enhance the interpretability and practical relevance of the results.

Include comparisons with other efficiency-oriented methods.
To better contextualize the contribution, the authors should compare LP against existing tensor-parallel or inference acceleration techniques (e.g., DeepSpeed-Inference, FasterTransformer, or attention–FFN fusion strategies). Such comparisons would clarify the unique advantages and limitations of LP.

---

> ### Author Response · Authors · 2025-11-16
> **Author response to Reviewer vws1**
>
> We thank the reviewer for their careful reading and constructive suggestions. In particular, we appreciate the request for a more principled analysis of the PAR approximation; the CKA-based experiment we added on Qwen3-4B has strengthened the empirical support for our layer-independence hypothesis.
>
> ## Requested changes
>
> ### Theoretical justification for the PAR approximation
>
> We have extended Section 3 with an additional representation-similarity analysis that probes how strongly attention and FFN sub-blocks depend on the residual stream from preceding layers. For prompts drawn from a small RedPajama subset, we record module activations under two conditions: a standard forward pass, and a counterfactual pass in which the incoming residual contribution is subtracted before the next block is evaluated. From these two runs we obtain, for each layer $k$, the original and counterfactual activations for both the attention and FFN submodules, and compute their CKA similarity scores $\mathrm{S}_A^k$ and $\mathrm{S}_F^k$. As shown in the new figure, layers that empirically tolerate the 2-parallel intervention exhibit high CKA similarity between original and counterfactual activations, indicating limited dependence of their attention and FFN computations on the immediate residual input. We emphasize in the text that this is an analytical, representation-level justification rather than a full theoretical derivation, and we explicitly leave a formal theory of layer decoupling to future work.
>
> ### Trade-off between accuracy and performance
>
> We have extended Table 2 by adding, immediately after “Eff. Depth,” a column reporting the relative speed-up over each model’s baseline. Timings are obtained by running forward passes on 100 RedPajama samples and averaging over three runs:
>
> | Model                   | Effective Depth | Speedup |
> | ----------------------- | --------------- | ------- |
> | Llama 2 7B (Chat)       | 32 (base)       | ×1.00   |
> | | 27              | ×1.15   |
> | | 26              | ×1.19   |
> | | 25              | ×1.23   |
> | | 24              | ×1.28   |
> | | 23              | ×1.31   |
> | Llama 3.2 3B (Instruct) | 28 (base)       | ×1.00   |
> | | 24              | ×1.12   |
> | | 23              | ×1.15   |
> | | 22              | ×1.19   |
> | | 21              | ×1.23   |
> | | 20              | ×1.28   |
> | Qwen3 4B (Instruct)     | 36 (base)       | ×1.00   |
> | | 31              | ×1.13   |
> | | 30              | ×1.15   |
> | | 29              | ×1.18   |
> | | 28              | ×1.21   |
> | | 27              | ×1.25   |
> | Qwen3 14B (Instruct)    | 40 (base)       | ×1.00   |
> | | 35              | ×1.12   |
> | | 34              | ×1.15   |
> | | 33              | ×1.18   |
> | | 32              | ×1.21   |
> | | 31              | ×1.24   |
> | | 30              | ×1.27   |
>
>
> Together with the accuracy columns already in Table 2, this provides an explicit speed–accuracy table. In the revised text we highlight representative operating points: for the larger models (Llama 2 7B and Qwen3 14B), LP without fine-tuning can deliver roughly 10% speed-up at the cost of only 0–3.75% average accuracy loss, which we view as a practically attractive regime. Conversely, we now explicitly characterize more aggressive settings---such as Qwen3-4B at effective depth 27, which yields ≈1.25× speed-up but a substantial accuracy drop---as exploratory configurations that probe the limits of layer parallelism rather than recommended operating points for deployment.
>
> ### Comparison to other tensor-parallel or efficiency methods
>
> We thank the reviewer for pointing us to recent work on communication-efficient tensor parallelism and have incorporated the suggested references into a new paragraph in the “Related work” section, where we discuss how LP relates to communication compression and sync-point reduction techniques for tensor-parallel inference.
>
> We already compare LP conceptually with attention–FFN fusion schemes used in models such as PaLM and GPT-J, emphasizing that LP operates at the level of the computational graph by altering the effective depth, rather than only fusing kernels. In the revision, we further clarify that systems such as DeepSpeed-Inference and FasterTransformer are inference engines that provide implementation-level optimizations (e.g., kernel fusion, scheduling, communication backends) that are, to the best of our knowledge, orthogonal to LP. We therefore frame direct empirical comparisons to those engines as out of scope for this work, while noting that LP could in principle be combined with such systems to obtain cumulative benefits.

---

### Decision · Action_Editor_Nn5j · 2025-12-21

**Recommendation:** Accept with minor revision

**Additional Comments:**

While the paper is in a good shape to be published, I agree with the reviewers that further improvement can be done via comparing to other existing methods for non-quality-neutral runtime optimization in terms of runtime/quality trade-off, and by providing more theoretical analysis/motivation of the presented selection of fused layers. Please try to incorporate these in the final revision (if no complete results, some discussions will also help).

**Audience:**

Yes

**Audience Explanation:**

This paper is about LLM runtime optimization, which is an important direction. The results are interesting to be shared with other researchers.

**Claims And Evidence:**

Yes

**Claims Explanation:**

This paper proposes layer parallelism to process consecutive layers pairs in parallel via tensor parallelism. The authors conducts extensive experiments on different settings with varying sizes and architectures. Experimental results consistently demonstrate the claims. The reviewers are mostly satisfied with the claims, excepts for some potential improvement specified below.

---

> ### Author Response · Authors · 2026-01-06
> **Reply to Action Editor Decision**
>
> Dear Action Editor and Reviewers,
>
> We sincerely thank the action editor and reviewers for the positive decision and for their constructive feedback, which has helped us improve the quality and clarity of our manuscript.
>
> We have carefully addressed the suggestions raised during the review process. In the revised version, we have included two new appendix sections:
>
> Appendix A: Theoretical Analysis of Layer Parallelism — This section provides a formal first-order error analysis of the LP approximation, deriving bounds on the error introduced by parallelizing consecutive layers. We connect these theoretical insights to our empirical observations, explaining why certain layers are more amenable to parallelization (via Jacobian sensitivity and residual magnitude arguments) and providing theoretical grounding for the layer selection strategy and the observed robustness patterns across model scales.
>
> Appendix B: Comparison with Other Tensor Parallelism Optimizations — This section presents a systematic comparison between Layer Parallelism and other recent non-quality-preserving methods for tensor-parallel inference optimization, including Sync-Point Drop (SPD), selective low-bit communication, and microscaling (MX) format compression. We provide quantitative comparisons of speedup and accuracy trade-offs across different model scales and hardware configurations, and discuss the complementary nature of these approaches.
>
> We believe these additions strengthen the contribution of our work and comprehensively address the reviewers' suggestions.
> Once again, we thank the reviewers for their time and valuable input.
>
> Sincerely,
> The Authors